# A New Ice Accretion Model for Aircraft Icing Based on Phase-Field Method

**Hao Dai [1], Chunling Zhu [1,\*], Huanyu Zhao [2] and Senyun Liu [3]**

1   College of Aerospace Engineering, Nanjing University of Aeronautics and Astronautics,
    Nanjing 210016, China; dai_hao_nuaa@163.com
2   Key Laboratory of Aircraft Icing and Ice Protection, AVIC Aerodynamics Research Institute,
    Shenyang 110034, China; zhao_huan_yu@163.com
3   Key Laboratory of Icing and Anti/De-Icing, China Aerodynamics Research and Development Center,
    Mianyang 621000, China; liusenyun@cardc.cn
*   Correspondence: clzhu@nuaa.edu.cn

**Abstract:** Aircraft icing presents a serious threat to the aerodynamic performance and safety of aircraft. The numerical simulation method for the accurate prediction of icing shape is an important method to evaluate icing hazards and develop aircraft icing protection systems. Referring to the phase-field method, a new ice accretion mathematical model is developed to predict the ice shape. The mass fraction of ice in the mixture is selected as the phase parameter, and the phase equation is established with a freezing coefficient. Meanwhile, the mixture thickness and temperature are determined by combining mass conservation and energy balance. Ice accretions are simulated under typical ice conditions, including rime ice, glaze ice and mixed ice, and the ice shape and its characteristics are analyzed and compared with those provided by experiments and LEWICE. The results show that the phase-field ice accretion model can predict the ice shape under different icing conditions, especially reflecting some main characteristics of glaze ice.

**Keywords:** numerical simulation; phase-field method; aircraft icing; ice accretion model

## 1. Introduction

The aircraft icing problem is one of the most serious problems in the aviation industry [1]. Aircraft ice accretion may not only increase the weight of aircraft, but also destroy the aerodynamic performance [2] and increase the resistance of aircraft [3], which is an important hidden danger leading to flight safety accidents. Consequently, in order to ensure the flight security of aircraft and the reliability of anti-icing and deicing systems, it is necessary to study ice accretion on aircraft surfaces. Aircraft icing is affected by many factors, including flight status, geometry of icing area, atmospheric conditions and duration of passing through icing clouds [4]. Among them, the characteristic parameters of atmospheric conditions and flight status are the most important external factors for ice accretion, including liquid water content (LWC), median volume diameter (MVD), ambient temperature, etc.

According to the ice shape, aircraft ice can be roughly categorized as rime ice, glaze ice and mixed ice [5]. Rime ice forms under the atmospheric conditions of low freezing temperature, usually below −15 °C, relatively low LWC and small MVD. After hitting the aircraft surface, the supercooled droplets freeze into ice rapidly, resulting in more gaps in the ice layer, less density and opacity, as seen from the appearance. Due to the instantaneous freezing characteristic, rime ice is loosely cemented and easily falls off, but its surface maintains an aerodynamic shape and is less harmful to the flight security of aircraft. Glaze ice mainly occurs when the ambient temperature is relatively high, usually in the range of −10 °C to 0 °C, with higher LWC and relatively larger MVD. In this case, supercooled droplets do not freeze or partially freeze after hitting the surface. A fraction

of residual liquid water flows under the action of external airflow and gradually freezes. The structure of glaze ice is relatively compact, with higher density and a transparent appearance, which does not easily fall off. In the process of ice accretion, horn ice may form on the leading edge of the wing, which has a great influence on the aerodynamic characteristics. Mixed ice is a mixture of rime ice and glaze ice, which has the characteristics of both ice types. Similar to glaze ice, mixed ice may also seriously damage the aerodynamic performance and pose a threat to flight safety.

At present, the approaches to studying aircraft ice accretion can be classified into two categories: experimental research with ground or flight tests [6–8] and numerical simulation methods [9,10]. In the process of research, each has its own advantages and disadvantages. In recent years, with the development of computer technology, the efficiency, economy and flexibility of numerical simulation methods are more and more prominent, and have been widely studied and applied. Since the 1940s, a number of icing simulation codes have been developed, such as LEWICE [11], ONERA [12], DRA [13], CIRAMIL [14] and FENSAP-ICE [15,16]. These icing simulation codes generally consist of three major modules, including calculating the airflow field, droplet trajectory and ice accretion.

In 1953, the Messinger model was first established based on energy balance [17]. Subsequently, Ruff [18] and Gent [19] improved the Messinger model by considering the temperature rise caused by compressible flow and adding an energy source at the substrate, which improved the applicability of the Messinger model. In order to build the Messinger model, the following limitations need to be adopted: the ice and water layers are isothermal. The heat is not allowed to be transferred from the interface of ice and water, and the energy can only be balanced by the latent heat generated during ice formation. Hence, Huang [20] introduced heat conduction in the ice layer into the classical Messinger model. These methods are relatively complex and difficult to implement in the icing code in a simple manner. However, these improvements have not changed the basic form of the Messinger Model. The Messinger model can accurately predict the ice shape of rime ice, but the prediction results for glaze ice and mixed ice need to be improved.

Bourgault [15] introduced the shallow water icing model (SWIM) into the icing calculation and established the differential equation of continuous water film flow on the icing surface. The SWIM can describe the morphology and distribution of liquid water on the ice surface in more detail, and improve the accuracy of ice prediction. Otta [21] assumed that the water film is controlled by a Couette flow driven by a shear force and the thickness of the water film is determined by the kinematic conditions at the interface between air and water, which together constitute the SWIM equations.

Meanwhile, Myers [22] extended the Messinger model for aircraft icing, involving solving heat equations in the ice and water layers. This method is very simple to introduce into the icing code. Cao [23] extended the heat transfer model of the Myers Model by introducing the concept of critical ice thickness as the criterion to judge whether there is overflow in each ice control volume. Thus, a simple and reasonable mathematical model was established, which can directly simulate the ice accretion on three-dimensional objects. Giulio [24] improved the Myers model based on the local, exact solution of the unsteady Stefan problem with heat transfer in the ice layer, and introduced local air temperature to replace the constant freestream value in the Myers model.

Aircraft icing is a very complex physical process, involving a phase change of the three-phase mixture of air, water and ice. Previous icing models focused on liquid water on the surface, or regarded ice and water as independent. In this study, we researched the flow and phase transition of a two-phase mixture of ice and water. The phase-field method is a powerful method to study this kind of multi-phase flow. The phase-field method treats multi-phase fluids as one fluid with variable material properties [25]. Specifically, the phase-field method utilizes order parameters such as the concentration or volume fraction of the fluids to represent different fluid phases. The order parameter is typically evolved by the Cahn–Hilliard (C-H) equation, the Allen–Cahn equation or other types of dynamic equations, so as to capture motion of the fluid interface.

In order to simulate the complex flows of a fluid mixture, especially binary incompressible fluid, a phase-field model was successfully established by Chella and Vinals [26], in which the total density and the density of each phase are constant in the model. In the same year, Gurtin, Poligone and Vinale extended the model to the framework of classical continuum mechanics and proved that it conformed to the second law of thermodynamics [27]. Antanovskii proposed a temperature-dependent free energy, and introduced the Cahn–Hilliard gradient term related to the phase-field model into the entropy function [28]. Recently, Guo proposed a numerical method for a quasi-incompressible Navier–Stokes–Cahn–Hilliard (NSCH) system that satisfied the laws of thermodynamics [29]. Du et al. [30] developed a numerical simulation method for the prediction of microstructural features of aircraft icing based on the developed phase-field method.

In this paper, we propose a new ice accretion model for aircraft icing based on the phase-field model. Different from previous icing models, in this study, we selected the mixture of ice and water as the research object, not just the continuous water film on the surface. Firstly, the icing problem of a two-dimensional wing is simplified. Then, based on the phase-field model, the water and ice are treated as one fluid, and the phase equation is established with the mass fraction of the mixture as the phase parameter. At the same time, according to the mass conservation and energy balance of the mixture, the mass conservation equation and energy conservation equation of the mixture are established and improved on the basis of the Messinger model and Myers model. With the establishment of the governing equations, the thickness and temperature of water film and ice layer can be determined so as to achieve the purpose of this study, to predict the aircraft ice shape under rime or glaze icing conditions.

The rest of the paper is organized as follows. In Section 2, we systematically establish and complete the governing equations of the new ice accretion model based on the phase-field method. In Section 3, the ice shapes are provided by the new ice accretion model under a series of icing conditions and compared with the experimental ice shapes and those provided by LEWICE. The findings are discussed and the future research directions are highlighted in Section 4. Finally, the main conclusions of the present work are provided in Section 5.

## 2. Materials and Methods

The numerical procedure of aircraft icing includes grid generation, flow field calculation, droplet trajectory calculation, ice accretion calculation, etc. The structured meshes generated by solving elliptic partial differential equations are taken as background grids, based on the given geometry. The Navier–Stokes equations based on the density pattern are solved to obtain the flow field. In the solver, the finite volume method based on a second-order central scheme with artificial viscosity by Jameson is applied for spacial discretization, and an explicit fourth-order Runge–Kutta algorithm is used for the time discretization. The S-A model is chosen for the turbulence model. In order to accelerate the convergence, the local time step and residual smoothing methods are introduced. The droplet motion equations based on Eulerian two-phase flow are solved to obtain the droplet trajectories. The spatial and temporal discretization methods are the same as the flow field equations. On this basis, the impingement characteristics of droplets can be calculated. Then, according to the droplet impingement characteristics, the ice shape can be predicted by the ice accretion model.

### 2.1. Simplification of Ice Accretion Model

In order to predict the ice shape under different ice conditions, it is necessary to build the control bodies on the icing area, and then build a mathematical model. This is different from the grid required for the calculation of airflow field and droplet trajectory, which is shown in Figure 1a. The control volume used in icing calculation is composed of nodes in a geometric shape. The control volume contains an ice layer and water film, as shown in Figure 1b. In this study, the ice and water in the control volume are treated as a

mixture, as shown in Figure 1c. $h$ is the average height of the mixture in the control volume, which is used as an unknown quantity in the icing calculation, but does not participate in the composition of the control volume. The upper surface of the control volume is in contact with the airflow, and the lower surface is the impact surface. The multi-layer grid is constructed as shown in Figure 1b.

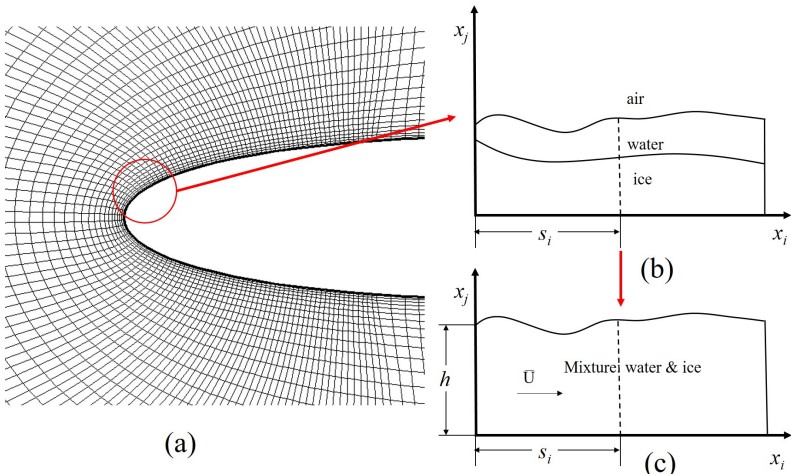

**Figure 1.** Diagram of ice model simplification: (**a**) the grid required for the calculation of airflow field and droplet trajectory; (**b**) the control volume of ice layer and water film; (**c**) the control volume of mixture.

As shown in Figure 1b, the chord length of mesh $s_i$ is much larger than its normal length $s_j$, $s_i \gg s_j$, and the normal velocity $u_j$ of the mixture can be neglected. Therefore, the multi-layer grid containing a mixture can be simplified to a single-layer grid with normal height of the mixture, which is shown in Figure 1c.

### 2.2. Two-Dimensional Form of Governing Equation

Aircraft icing is a special fluid–solid phase change phenomenon. However, both the Messinger model and the shallow water film model only study the continuous water film on the surface. In the phase-field icing model, the mixture of water and ice is taken as the research object. Therefore, the phase equation, the mass equation and the energy conservation equation of the mixture need to be established. By solving these governing equations, the mixture thickness and the mass fraction of ice in each grid are predicted. On this basis, the ice thickness and the ice shape can be obtained.

In the control volume $V(t)$, the mass of the mixture, water phase and ice phase are denoted by $M$, $M_w$ and $M_i$, respectively. Then, we obtain [25]:

$$M = M_w + M_i, \rho V = \rho_w V_w + \rho_i V_i., \tag{1}$$

Among them, $V$, $V_w$ and $V_i$ are the total volume of mixture, the volume of water phase and the volume of ice phase in the control volume, respectively. Their expressions are as follows:

$$V = s \cdot h, V_w = s \cdot h_w, V_i = s \cdot h_i, \tag{2}$$

where $h$ is the average normal height of the mixture over the control volume, $h_w$ is the average normal height of water phase and $h_i$ is the average normal height of ice phase, $h = h_w + h_i$.

By combining Equations (1) and (2), we can obtain:

$$\rho h = \rho_w h_w + \rho_i h_i, \tag{3}$$

and

$$\rho h \mathbf{U} = \rho_w h_w \mathbf{U}_w + \rho_i h_i \mathbf{U}_i, \tag{4}$$

where $\mathbf{U}$ is the velocity of the mixture, $\mathbf{U}_w$ is the velocity of the water phase and $\mathbf{U}_i$ is the velocity of the ice phase.

In this study, the ratio of ice mass to mixture mass is chosen to be the phase parameter $C$, which means $C = (\rho_i h_i)/(\rho h)$, $\rho_w h_w = 1 - C$. We obtain:

$$\mathbf{U} = (1 - C)\mathbf{U}_w + C\mathbf{U}_i. \tag{5}$$

Assuming that the ice produced by freezing adheres to the surface without displacement, the average velocity is as follows:

$$\mathbf{U}_i = 0. \tag{6}$$

Then, the velocity of the mixture can be described as:

$$\mathbf{U} = (1 - C)\mathbf{U}_w. \tag{7}$$

The conservative equation of phase parameter $C$ can be written as:

$$\frac{\partial \rho h C}{\partial t} + \nabla \cdot (\rho h C \mathbf{U}) = \dot{m}_{ice}. \tag{8}$$

In the phase equation, $m_{ice}$ is the amount of liquid water frozen into ice, which can be described as:

$$m_{ice} = f_{ice} \cdot (1 - C)\rho h, \tag{9}$$

where $f_{ice}$ is the freezing rate, and can be written as [31]:

$$f_{ice} = \frac{Cp_w}{Lf} \cdot (T_w - T_\infty - \frac{u_d^2}{2Cp_w} + \frac{\phi \cdot htc}{Cp_w V_\infty LWC\beta}), \tag{10}$$

$$\phi = -T_\infty - 0.89 \cdot \frac{u_d^2}{2Cp_w} + \frac{\lambda \vartheta Le}{Cp_a htc}, \tag{11}$$

$$\vartheta = (\frac{e_s(T_w)}{T_\infty} - \frac{P_{total}e_s(T_\infty)}{P_\infty T_{total}})/(\frac{P_{total}}{0.622 T_{total}} - \frac{e_s(T_w)}{T_\infty}), \tag{12}$$

in which $htc$ is the convective heat transfer coefficient. $T_w$ is the temperature of the water film. $Cp_w$ is the specific heat capacity and water. $T_m$ is the the phase transition equilibrium temperature of water, $T_m = 273.15$ K. $Le$ and $Lf$ are the latent heat of water gasification and ice condensation. $u_d$ is the velocity of a droplet impacting on the surface. $\lambda$ is thermal conductivity of air. $T_\infty$ and $P_\infty$ are the temperature and pressure of the incoming stream. $T_{total}$ is the total temperature of the incoming stream and $P_{total}$ is the total pressure of the incoming stream, which can be specified as follows [18]:

$$T_{total} = T_\infty \cdot (1 + 0.2 \cdot Ma^2), \tag{13}$$

$$P_{total} = P_\infty \cdot (1 + 0.2 \cdot Ma^2)^{\frac{\gamma}{\gamma - 1}}. \tag{14}$$

As shown in Figure 2, the conservation equation of mass for the mixture can be described as:

$$\frac{\partial \rho h}{\partial t} + \nabla \cdot (\rho h \mathbf{U}) = \dot{m}_{imp} - \dot{m}_{evp}, \tag{15}$$

in which $\dot{m}_{imp}$ is the water droplet collection rate. $\dot{m}_{evp}$ is the evaporation rate, which is obtained by analogy with the convection heat transfer term [17],

$$\dot{m}_{imp} = LWC \cdot V_\infty \cdot \beta, \dot{m}_{evp} = \frac{htc}{Cp_a \rho_a R_v Le w^{2/3}}[\frac{e_s(T_w)}{T_w} - \frac{e_s(T_\infty)}{T_\infty}]. \tag{16}$$

Among them, $Cp_a$ is the specific heat capacity of air at constant pressure, which can be taken as 1003.5 J/(kg · K); $\rho_a$ is the air density; $R_v$ is the gas constant of water vapor, which can be taken as 461 J/(kg · K); *Lew* is the analogy criterion number of heat and mass transfer, which is related to the content of liquid water in the air, and can be taken as 1 in general icing conditions; $e_s(T_w)$ and $e_s(T_\infty)$ are the saturated steam pressure corresponding to the water temperature $T_w$ of the control volume and the ambient temperature $T_\infty$, respectively [20].

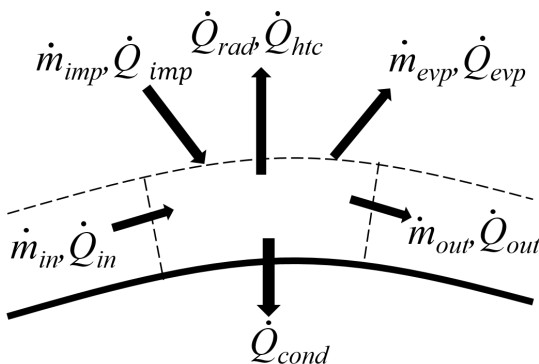

**Figure 2.** Diagram of mass and energy transfer in the control volume.

When $T_w \geq T_m$,

$$e_s(T_w) = 611.011 + 44.481(T_w - T_m) + 1.419(T_w - T_m)^2$$
$$= +0.0239(T_w - T_m)^3 + 1.744 \times 10^{-4}(T_w - T_m)^4, \tag{17}$$

and when $T_w < T_m$,

$$e_s(T_w) = 609.603 + 49.495(T_w - T_m) + 1.739(T_w - T_m)^2$$
$$= +0.031(T_w - T_m)^3 + 2.292 \times 10^{-4}(T_w - T_m)^4, \tag{18}$$

$T_m$ is the the phase transition equilibrium temperature of water, $T_m = 273.15$ K.

The energy equation is given by:

$$\frac{\partial(\rho h H)}{\partial t} + \nabla \cdot (\rho h H \mathbf{U}) = \dot{Q}_{rad} + \dot{Q}_{imp} + \dot{Q}_{evp} + \dot{Q}_{htc} + \dot{Q}_{cond}. \tag{19}$$

Among them, $H$ is the internal energy of the mixture, which can be obtained as:

$$H = CpT + (1-C)L = (1-C)Cp_w T_w + CCp_i T_m + (1-C)Lf, \tag{20}$$

where $Cp_i$ is the specific heat capacity of ice and water, respectively. $\dot{Q}_{rad}$ is the radiation heat flux of the water film. $\dot{Q}_{imp}$ is the collected water energy and includes the internal energy and kinetic energy of the droplet. $\dot{Q}_{evp}$ is the energy density taken away by evaporation or sublimation on the surface of the water film. $\dot{Q}_{htc}$ is the convective heat transfer density between the water film and the air flow. $\dot{Q}_{cond}$ is the energy transmitted by the aircraft skin. The unknown terms are specified in the form of [22]:

$$\dot{Q}_{rad} = \sigma_r(T_w^4 - T_\infty^4), \tag{21}$$

$$\dot{Q}_{imp} = \dot{m}_{imp}[Cp_w(T_w - T_\infty) - \frac{u_d^2}{2}], \tag{22}$$

$$\dot{Q}_{evp} = \dot{m}_{evp} Le, \tag{23}$$

$$\dot{Q}_{htc} = htc(T_w - T_{rec}). \tag{24}$$

As shown in Figure 3, the water film flow can be simplified as a Couette flow under the action of air shear stress, and the velocity of the water film can be approximately described in the form of a linear distribution [21]:

$$\mathbf{U}_w(s,y) = \frac{y}{\mu_w}\tau_a(s),\tag{25}$$

where $s$ is the arc direction of the surface and $y$ is the normal direction of the surface. Then, the average flow rate of the water film can be defined as:

$$\overline{\mathbf{U}}_w(s,y) = \frac{h_w}{\mu_w}\tau_a(s),\tag{26}$$

in which $\tau_a(s)$ is the shear stress of airflow on the water film.

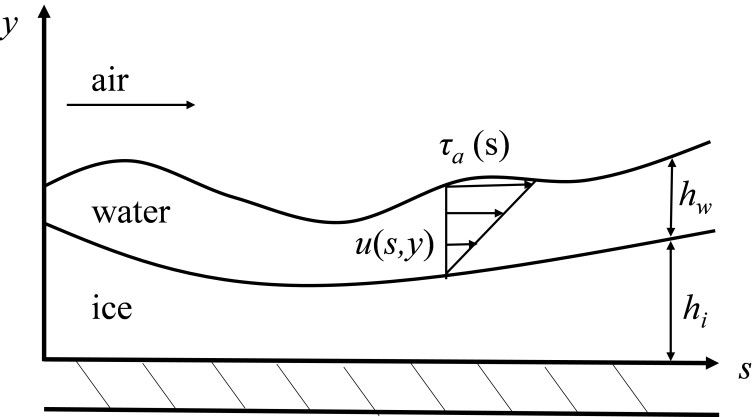

**Figure 3.** Diagram of water film and ice.

In icing simulations, the convective heat transfer process *htc* has a great influence on the ice shape, and is the key to solving the convective heat transfer process. At present, most icing simulations use the boundary layer integral functions proposed by LEWICE, which consider surface roughness and velocity variation to solve the convection heat transfer coefficient [11].

$$htc = \begin{cases} 0.296 \cdot \frac{\lambda_s}{v_a^{0.5}}(V_e^{-2.87}\int_0^s V_e^{1.87}ds)^{-0.5}, & \mathrm{Re}_k < \mathrm{Re}_c \\ \frac{C_f/2}{Prt+\sqrt{C_f/St_k}} \cdot \rho_a \cdot V_e \cdot Cp_a, & \mathrm{Re}_k \geq \mathrm{Re}_c \end{cases}\tag{27}$$

where $\lambda_s$ is the thermal conductivity of air. $v_a$ is the kinematic viscosity of air. $V_e$ is the velocity of air at the outer boundary of the boundary layer. $s$ is the surface distance. $Prt$ is a Prandtl number, which is approximated as a constant 0.9. $\mathrm{Re}_k$ and $\mathrm{Re}_c$ are, respectively, the Reynolds number at the rough surface and the critical Reynolds number. $C_f$ is the friction coefficient between the air flow and the wall surface. $St_k$ is a rough Stanton number, which can be obtained by

$$St_k = 0.8 \cdot [\frac{\cdot Ve \cdot k_s C_f}{2v_a}]^{-0.2}Prt^{-0.44},\tag{28}$$

where $k_s$ is the roughness height.

The solution process of the phase-field ice accretion model is as follows: the liquid water temperature $T_w$ is solved by the energy balance equation, and the freezing coefficient is determined by the liquid water temperature $T_w$, so as to solve the phase equation and the mass conservation equation. The final ice thickness and the mass fraction of ice are obtained by iteration.

## 3. Results

In order to compare the shapes and the ice shape characteristics chosen, provided by experiments, LEWICE and the phase-field icing model, a total of eight ice shapes and the icing conditions are selected from the literature [32,33], which are listed in Table 1. For the purpose of comparison, the stagnation thickness and horn angle are chosen to be the ice shape characteristics, which are determined for the reference and prediction ice shapes. Through these characteristic values illustrated below in Figure 4, it is easy to calculate the deviations between the experimental shapes and the prediction shapes provided by different icing codes.

Before comparing the ice shapes and the ice shape characteristics, a study on grid independence is needed first. In this study, Case 1 and Case 4 are selected as icing conditions, and 80 nodes, 120 nodes and 200 nodes are arranged in the icing area of the wing leading edge. Wing leading edge icing and its node arrangement are shown in Figure 5. The calculation results are shown in Figure 6. As can be seen from the figures, Case 1 is a typical rime ice and Case 4 is a typical glaze ice. The results show that in the case of rime ice, the results of the three node layout schemes are similar. While in the case of glaze ice, the ice shapes provided by 120 nodes and 200 nodes are similar, 80 nodes cannot accurately describe the glaze ice shape. Therefore, for the sake of accuracy and efficiency, the verification calculation in this paper adopts the 120-node layout method.

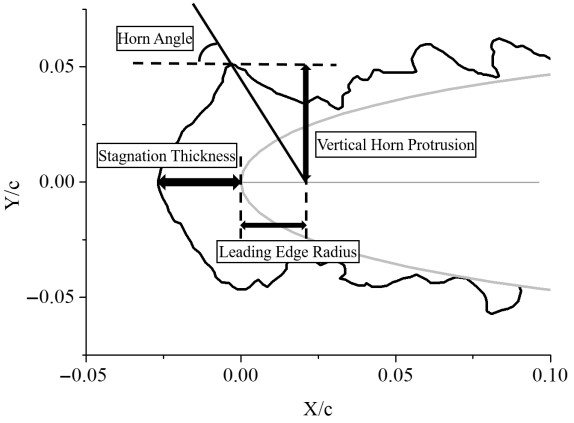

**Figure 4.** Ice shape characterization metrics.

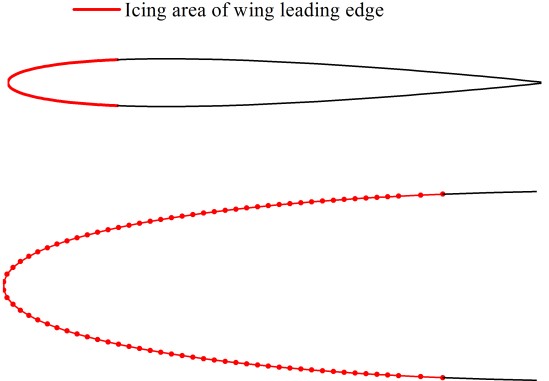

**Figure 5.** Wing leading edge icing and its node arrangement.

**Table 1.** Icing conditions for verification cases.

| Verification Case | Pressure Pa | MVD μm | Chord m | AOA deg | Mach Number | LWC g/m³ | Temperature K | Time s |
|---|---|---|---|---|---|---|---|---|
| Case 1 | 101,325 | 20 | 0.5334 | 4 | 0.197 | 1.0 | 244.8 | 360 |
| Case 2 | 101,325 | 20 | 0.5334 | 4 | 0.197 | 0.5 | 260 | 360 |
| Case 3 | 101,325 | 20 | 0.5334 | 4 | 0.197 | 1.0 | 262.05 | 360 |
| Case 4 | 101,325 | 20 | 0.5334 | 4 | 0.197 | 1.0 | 267.55 | 360 |
| Case 5 | 101,325 | 19 | 0.3531 | 0 | 0.292 | 0.75 | 263.25 | 600 |
| Case 6 | 101,325 | 30 | 0.5334 | 4 | 0.313 | 1.8 | 263.07 | 360 |
| Case 7 | 101,325 | 20 | 0.5334 | 4 | 0.207 | 1.0 | 259.85 | 360 |
| Case 8 | 101,325 | 20 | 0.5334 | 0 | 0.178 | 2.1 | 263.45 | 300 |

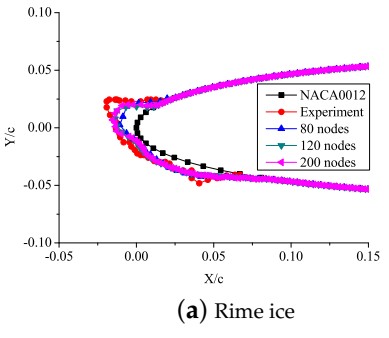
(**a**) Rime ice

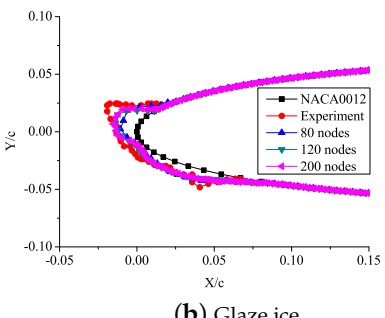
(**b**) Glaze ice

**Figure 6.** Grid independence test.

Figure 7 shows the distribution of water film thickness under four cases, from Case 1 to Case 4, calculated by the phase-field model. It can be seen that when the ambient temperature is relatively low, the icing type is rime ice. At this time, the water film thickness is almost 0, and the droplets freeze immediately when hitting the surface. When the icing temperature is high, there will be water film distribution on the surface. The water film thickness at the leading edge is larger, and the more it flows downstream, the thinner the water film is. This is due to the large amount of water droplets collected at the leading edge and the small surface convective heat transfer coefficient. With the flow of the water film, the liquid water gradually freezes into ice, so the water film gradually decreases. At the same time, it can be seen from the figure that with the increase in freezing temperature, the water film thickness increases gradually. This is because the higher the temperature is, the lower the freezing coefficient of water drops is, and more liquid water forms on the surface, resulting in the increase in water film thickness.

Figure 8a shows the comparison of icing shapes in the case of rime ice. It can be seen that the ice shapes can be well simulated by both LEWICE and the phase-field model in this study. The freezing temperature is relatively low at this time, and the droplet freezes immediately after impacting on the surface. The liquid water on the surface does not flow, so the simulation accuracy is relatively high. Meanwhile, the stagnation thickness from the phase-field model is closer to that of the experiment than that from LEWICE.

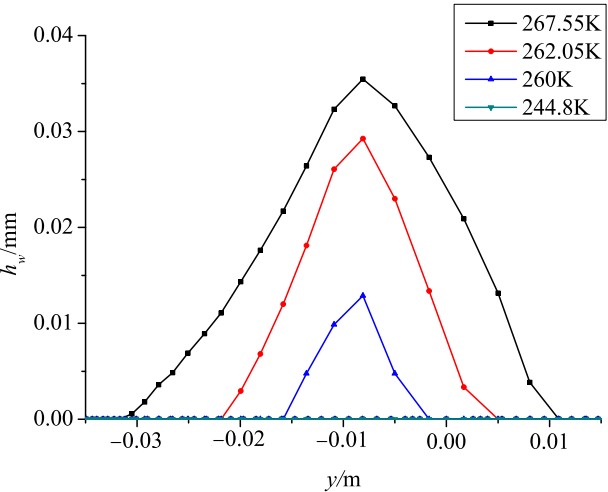

**Figure 7.** Distribution of water film thickness on the surface.

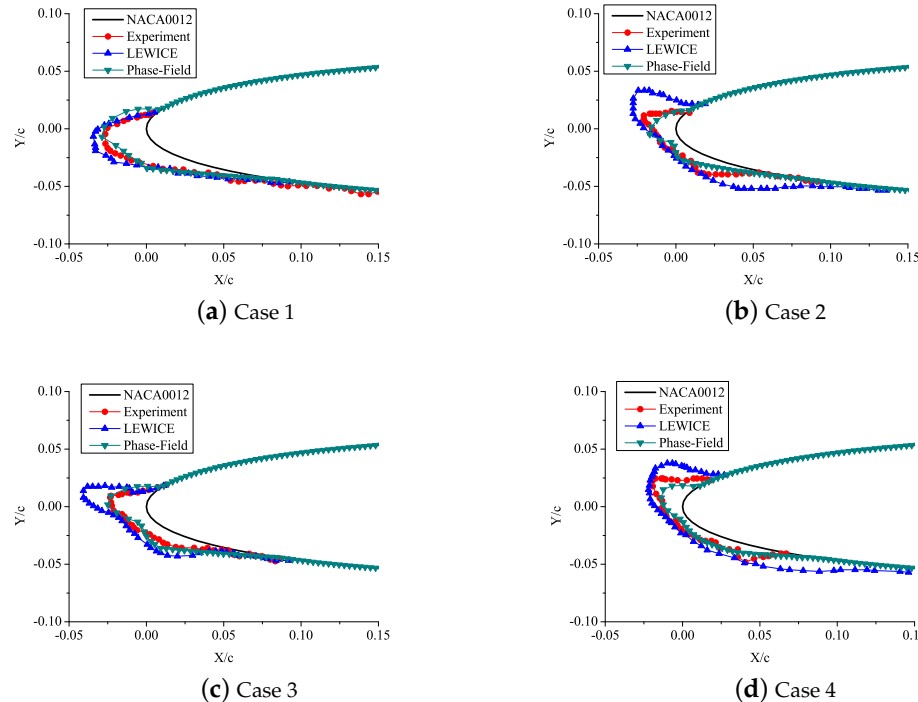

**Figure 8.** Comparison of the ice shapes under different verification cases.

Figure 8b,c show the comparison of ice shapes when the incoming temperature is between −11 °C and −15 °C, which can be considered as mixed ice. It can be seen from the figures that the ice shapes simulated by LEWICE still have a large deviation. The simulation results of the phase-field model are closer to the experimental values, especially the prediction of the top ice horn. Therefore, the accuracy of the phase-field model is higher than that of LEWICE under this freezing weather condition.

The ice shape provided in Figure 8d is a typical glaze ice. It can be seen that the simulation accuracy of the ice shape from LEWICE is relatively poor, and the shapes of the upper and lower surfaces are very different from the experimental values. Meanwhile, the ice shape predicted by the phase-field model is more similar to the experimental ice shape than the other icing code.

As shown in Table 2, the deviation of stagnation thickness provided by the phase-field model and the experiment is smaller than that by LEWICE.

The above comparison verifies the effectiveness and accuracy of the phase-field icing model under different temperatures. Then, the accuracy and effectiveness of the phase-field icing model under other icing weather conditions are verified, including different incoming flow velocity, different LWC and different MVD.

**Table 2.** Stagnation thickness of ice shapes provided by different icing codes and experiment.

|  | Case 1 |  | Case 2 |  | Case 3 |  | Case 4 |  |
|---|---|---|---|---|---|---|---|---|
| Experiment | 0.02518 |  | 0.0155 |  | 0.0223 |  | 0.01319 |  |
| Phase field | 0.02717 | 7.9% | 0.01579 | 1.9% | 0.02493 | 11..8% | 0.01246 | −5.5% |
| LEWICE | 0.03148 | 24.8% | 0.02092 | 35.0% | 0.03419 | 53.3% | 0.01813 | 37.5% |

Figure 9 show the comparison of ice shapes from the experiment, phase-field model and LEWICE under different icing conditions. Table 3 shows that the deviation of stagnation thickness provided by the phase-field model and the experiment is also smaller than LEWICE. This set of data and figures demonstrate that the ice shapes from the phase-field model are closer to the experimental shapes than those from LEWICE.

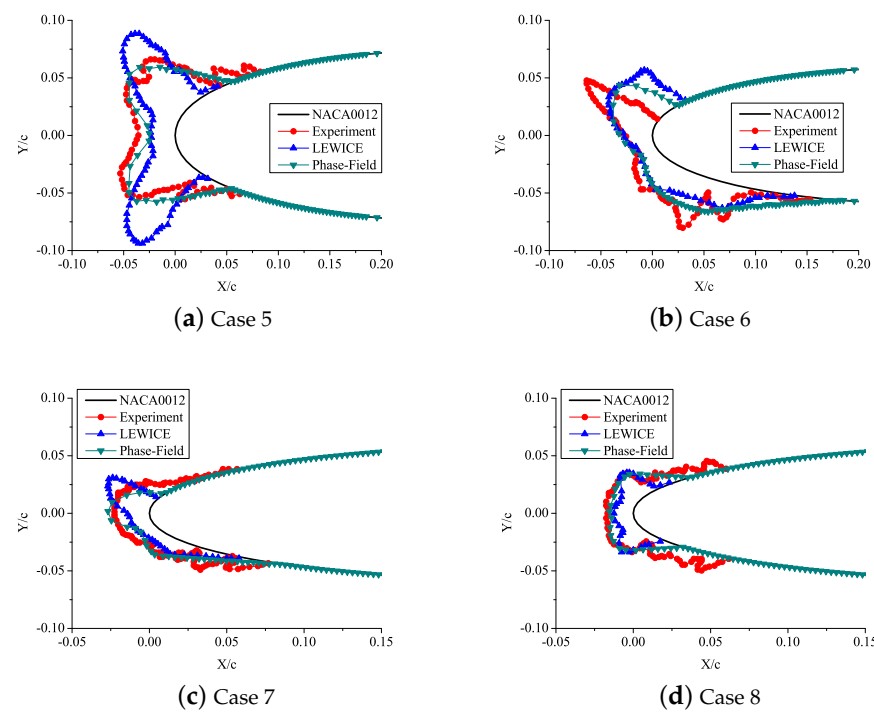

**Figure 9.** Comparison of the ice shapes in verification cases from Case 5 to Case 8.

**Table 3.** Stagnation thickness of ice shapes provided from Case 5 to Case 8.

|  | Case 5 |  | Case 6 |  | Case 7 |  | Case 8 |  |
|---|---|---|---|---|---|---|---|---|
| Experiment | 0.03571 |  | 0.255 |  | 0.02256 |  | 0.01667 |  |
| Phase field | 0.02501 | −30.0% | 0.02632 | 3.2% | 0.02679 | 18.8% | 0.01536 | −7.9% |
| LEWICE | 0.02302 | −35.5% | 0.02709 | 6.2% | 0.01402 | −37.9% | 0.0125 | −25.0% |

From the simulation of the above eight states, it can be seen that the phase-field icing model proposed in this paper can simulate the icing shape under various conditions, and

the simulation results are closer to the experimental values, which better demonstrate the correctness of the model.

## 4. Discussion

Within the framework of this study, the ability of the phase-field method in aircraft icing calculation was confirmed, and a phase-field icing model and its icing code was developed. Like normal phase-field models, the phase-field icing model treats multi-phase fluids as one fluid with variable material properties.

At present, ice accretion models are mainly divided into the Messinger icing model and shallow water film icing model. In the Messinger icing model, the mass and energy conservation equations of the control volume are established based on the surface liquid water to solve the equilibrium temperature of the control volume, and then the amount of ice accretion is calculated. The shallow water film icing model is based on the distribution of liquid water on the surface, and the ice accretion model is established based on the movement of the water film. The ice shape predicted for glaze ice and mixed ice needs to be improved. Compared with the two kinds of ice accretion models above, the phase-field icing model takes an ice–water mixture as the research object, and establishes the ice accretion model based on the phase-field method.

The flight state and weather conditions of the aircraft will affect the ice shape and the freezing level. The flight state includes flight speed and angle of attack (AOA). The meteorological conditions include the ambient temperature, liquid water content (LWC), mean volumetric diameters (MVD) and ice time. In order to verify the accuracy and applicability of the phase-field icing model, eight different icing conditions are selected, in which the incoming Mach number is from 0.197 to 0.313, the LWC is from 0.5 $g/m^3$ to 2.1 $g/m^3$, the incoming temperature is from 244.8 K to 267.55 K, the icing time is from 300 s to 600 s, the AOA is from 0° to 4° and the MVD is from 19 μm to 30 μm. The predicted ice types include rime ice, glaze ice and mixed ice. For any icing condition, the ice shape and its characteristic points are more consistent with the experimental ice shape than those provided by LEWICE.

Although the correctness and effectiveness of the phase-field icing model to predict the two-dimensional icing ice shape under different icing conditions have been verified, the correctness of the model when predicting the three-dimensional icing shape needs to be developed further. At the same time, the current phase-field icing model is only a two-dimensional phase-field problem, which can be developed into a three-phase problem of ice water and air in the future.

## 5. Conclusions

In this paper, a new numerical simulation method for predicting ice shape is proposed. The ice and water in this study are regarded as one fluid, and the method is implemented based on the phase-field model. The main advantage of the phase-field model is predicting the ice shapes under glaze ice or mixed ice conditions, which mean the ambient temperature is high. By comparing three node layout schemes under different icing conditions, the sensitivity of the mesh was verified. Then, comparing the simulation results, the ice shapes predicted by the phase-field icing model are found to be more consistent with the experimental ice shapes than those predicted by LEWICE, especially the mixed ice and the glaze ice. Taking the stagnation thickness as the ice shape characteristic, the deviation between the ice shape provided by the phase-field model and experimental ice shape is the smallest. Therefore, the validity and accuracy of the ice accretion model based on field phase method is verified.

**Author Contributions:** Conceptualization, H.D., C.Z., H.Z. and S.L.; methodology, H.D.; software, H.D.; validation, H.D.; formal analysis, H.D.; investigation, H.D.; resources, H.D.; data curation, H.D.; writing—original draft preparation, H.D.; writing—review and editing, H.D. and C.Z.; visualization, H.D.; supervision, C.Z.; project administration, C.Z.; funding acquisition, C.Z. and S.L. All authors have read and agreed to the published version of the manuscript.

**Funding:** This research was funded by the National Natural Science Foundation of China (Grant No.11832012, 51806105), Open Fund of the Key Laboratory of Icing and Anti/De-icing (Grant No. IADL20190302, IADL20190309) and the Natural Science Foundation of Jiangsu Province (Grant No. BK20180442).

**Institutional Review Board Statement:** Not applicable.

**Informed Consent Statement:** Not applicable.

**Data Availability Statement:** The data presented in this study are available on request from the corresponding author.

**Conflicts of Interest:** The authors declare no conflict of interest.

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
