# Peer review of "A New Ice Accretion Model for Aircraft Icing Based on Phase-Field Method"

_applsci, doi:10.3390/app11125693_

Round 1
Reviewer 1 Report
This paper entitled "A new Ice Accretion Model for Aircraft Icing Based on Phase-field Method" presents a mathematical model to predict ice accretion on aircraft. This model is based on the phase-field method in which ice and liquid water is treated as one fluid. The Messinger model and its extension using Myers’ heat-transfer extension are used to construct the model.
The mass fraction of ice in the mixture is selected as the phase parameter. The proposed model is written using partial differential equations expressing mass and energy conservation.
The paper is organized as follows: an introduction in the first section in which the authors position their work according to literature. Section 2 introduces the governing equations based on the phase-field method. Section 3 describes some test cases to verify the model. Ice shapes are compared to experimental and numerical shapes. Section 4 deals with discussion and future research directions. And section 5 is the conclusion.
In the abstract, the sentence, line 3, mentioning Messinger and Myers works should be rephrased. The last sentence of the abstract is not specific enough and does not provide any precise information.
In the introduction, line 32, the aerodynamic surface is not smooth but rough in case of rime ice accretion but the shape remains aerodynamic.
In section 2, the authors should clarify what they call control bodies. Is it a control volume or a grid? The authors mention, as shown in figure one, that this control body is different from the grid. The authors have to be more specific about that and explain $s_i$ and $s_j$ according to the referent grid.
In equation (2), is $h$ an average value over the control volume i.e. a constant?
In equation (6), the authors speak about the average velocity equals to 0 but then they assume that the velocity $U_i$ is also 0. The authors should clarify their notation.
Before equation (9) there are two commas in the sentence.
In equation (10) $C_{pw}$ is not defined, like $T_m$ which is defined later.
Before equation (21), a point is missing before “The unknown”.
Equation (21), there is a sign mistake.
Also in equation (22) \dot{Q}_{imp}=\dot{m}_imp*[C_{p,w}*(T_{d,\infty} – T_m) +u^2_d/2].
Page 6, line 162: Couette flow not Coutte flow.
Page 7, the first sentence of the Results section should be rephrased. The authors should also justify the ice shape characteristics chosen.
Page 8, Figure 6, a lot of information is missing. Lewice calculations are single-layered or multi-layered? Since the authors want to verify their icing model they should compare first the collection efficiency and the convective heat transfer coefficient. They should also compare ice shape using exactly the same collection efficiency and convective heat transfer coefficient as LEWICE to focus on the icing model only.
Authors should also add a sensibility study to the mesh to see how the model behaves when the mesh is refined.
The discussion of the results and the conclusion should be revised according to the additions proposed previously.
On the whole, I recommend the authors to revise the paper in accordance with the above comments.
Best Regards.
Author Response
Dear Reviewer:
Thank you very much for the comments concerning our manuscript “A New Ice Accretion Model for Aircraft Icing Based on Phase-Field Method” (ID: applsci-1234367). Those comments are all valuable and very helpful for revising and improving our paper, as well as the important guiding significance to our researches. We have looked through the comments and have made careful corrections. The revised portions are marked in red in the paper.
We are extremely grateful to the reviewer for the valuable suggestions and comments. According to the suggestions and comments put forward by the reviewer, detailed explanations and modifications are made in appendix.

Reviewer 2 Report
This study focuses on modelling aircraft ice modelling using phase-field method what is argued to a less-researched field in aircraft ice modelling. It is well-written. The results are compared with some experimental results available in the literature. The topic is interesting and relevant to the journal’s scope. There are some general and specific comments listed below.
- There are too many details about the methodology in the abstract. You could also mention the importance of the problem, the context and scope of the study and the application area in the abstract.
- In line 82 of page 2 the authors state that the novelty of the crunch paper is to implement a face field method for studying the aircraft icing could you please describe what is the difference of this study and other studies conducted by other researchers implementing the phase field method for aircraft icing such as: https://link.springer.com/article/10.1007/s10765-019-2585-2
- section 2.2 drives different equations in a step-by-step approach. However, this section could also be improved by adding some fundamental discussions from the viewpoint of ice accretion. For example, discussing what the purpose of section 2.2 is and what is going to be modelled, what the model output is would be very much appreciated. The model output should be clearly mentioned and described. After section 2.2 the authors could also briefly discuss how the models can be solved numerically and implemented.
- Could others develop a table of notation?
- What is the reason of comparing the results against only one simulation method LEWICE and not the other approaches?
- What is the reason behind choosing a one-dimensional problem? could you please shortly discuss the modelling framework for a two- and three-dimensional problems in terms of possible required modifications.
- There are a lot of equations on pages 4-6. However, there is no reference and that makes the reader think if all is the questions are derived and suggest it by the authors while many of these equations are famous and known expressions in the field of thermodynamics and fluid mechanics. Could you please use some references whenever appropriate and please be clear if there are questions that you have driven as a part of this research in other words the others should see if it is necessary to have all the steps of developing the equations. It is not clear to understand the authors original contribution in this set of equations.
- The study compares the models results with some experimental results. Could you elaborate on experimental results? Are they obtained by the authors? If so, the authors should have a clear description of the experiment setups, etc. If the experiment results are taken from other studies, a brief discussion on their experimental results is very much appreciated.
Some minor comments:
- Abstract line 1: What is meant by simply and directly?
- Abstract line 3: It should read “In combination with …”.
- Page 2 line 36 should read “a fraction of residual liquid water flows …”.
- Page 2 line 57 it should read: the heat is not allowed
- The second Paragraph of page 2 is too large. Could you break it down into several paragraphs as there are different concepts described in that large paragraph?
- Page 2 line 56 what is meant by “to ensure the deduction”?
- Line 93 Page 3: Could you please not start a sentence with “and”?
- Please define parameter U in equation 4
- Lines 141 and 948 should start with small letters not capital ones. The same comment applies to other parts of the paper whenever necessary for instance the paragraph after equation 12 and line 149, the first line after a question 20, the first line should not be indented and should start with a small letter, or the paragraph after equation 25 and line 165,
Author Response

(The authors gave the same response as above.)

Reviewer 3 Report
Icing simulation is a very important topic in aviation, and many readers would be interested in this paper. In this paper, the authors proposed a new icing model and indicated that their model can well predict ice shapes under various icing conditions, comparing with the results obtained by the NASA LEWIS code. However, following points should be revised for publication.
(1) Heat transfer coefficient htc is the key parameter in icing simulations. The authors should describe the model used in this study.
(2) Numerical procedure is not written at all. Grid, algorithm, numerical scheme, turbulence model, grid-independence study and so on should be expressed in one chapter.
(End)
Author Response

(The authors gave the same response as above.)

Round 2
Reviewer 1 Report
Dear Authors,
Thank you for your revised manuscript.
I still have one concern about the convective heat transfer coefficient.
The authors mention, page 4, RANS simulations using Spalart-Allmaras turbulence model but then in page 8, for $htc$ they mention boundary layer integral functions. So it is not clear to me how the authors compute $htc$. This part should be explained and the computed $htc$ should be compared to LEWICE convective heat transfert coefficient.
Best Regards
Author Response
Dear Reviewer:
Thank you very much for the comments concerning our manuscript “A New Ice Accretion Model for Aircraft Icing Based on Phase-Field Method” (ID: applsci-1234367). Those comments are all valuable and very helpful for revising and improving our paper, as well as the important guiding significance to our researches. We have looked through the comments and have made careful corrections. The revised portions are marked in red in the paper.
We are extremely grateful to the reviewer for the valuable suggestions and comments. According to the suggestions and comments put forward by the reviewer, detailed explanations and modifications are made as follows:
Comment 1: The authors mention, page 4, RANS simulations using Spalart-Allmaras turbulence model but then in page 8, for $htc$ they mention boundary layer integral functions. So it is not clear to me how the authors compute $htc$. This part should be explained and the computed $htc$ should be compared to LEWICE convective heat transfert coefficient.
Response: Thank you for your comments. In the calculation of flow field, the boundary layer can be obtained by turbulence model, the $htc$ is greatly affected by the grid size. Therefore, the boundary layer is not directly used in the calculation of convective heat transfer coefficient. LEWICE proposed a boundary layer integral functions, which is widely used in icing numerical simulation. The advantage of the boundary layer integration method is that it does not need to rely on the flow field grid. In this paper, $htc$ is calculated by the same empirical formula (Line 237) proposed by LEWICE. So the result of convective heat transfer coefficient is the same, as shown in the figure below.
I'm sorry for the misunderstanding caused by the imprecise expression, and have modified the expression in the manuscript in Line 236.
At present, most icing simulations use the boundary layer integral functions proposed by LEWICE, which consider surface roughness and velocity variation to solve the convection heat transfer coefficient.

Reviewer 2 Report
I have no further comments. Thank you for reflecting on the comments.
Author Response
Dear Reviewer:
Thank you very much for the comments concerning our manuscript “A New Ice Accretion Model for Aircraft Icing Based on Phase-Field Method” (ID: applsci-1234367).